# Research on High-Speed Railway Pricing and Financial Sustainability

**Xiaoyi Hu, Jianqiang Duan * and Ran Li**

School of Economics and Management, Beijing Jiaotong University, Beijing 100044, China;
17113171@bjtu.edu.cn (X.H.); li.ran@bjtu.edu.cn (R.L.)
* Correspondence: jqduan@bjtu.edu.cn

**Abstract:** With the steady increase of corporate system reform in the railway transportation industry, high-speed railways have accelerated their steps toward marketization, and the market competition they face has become increasingly fierce. In this context, enterprises need to make quick adjustments to new changes in order to seize the market share, obtain stable ticket income, and eventually, achieve financial sustainability while ensuring the healthy development of high-speed railways. Thus, it is particularly important to determine a reasonable ticket price. While considering supply and demand, market competition, and passenger utility, this study constructs a model of the passenger flow sharing rate and applies it to modeling the optimal ticket price for high-speed railways, taking the Beijing–Shanghai High-speed Railway as an example for calculating this model. Using the results calculated, this study analyzes the influence of the optimal ticket price on the financial sustainability of the enterprise through use of the evaluation system of financial sustainability established above. The results show that the existing price for a train ticket from of Beijing to Shanghai is not the optimal one, and there is still room for a price increase; the ticket price has an impact on financial sustainability by affecting corporate operating income and cash flow. The study provides a method for formulating the optimal ticket price for high-speed railway travel and enriches the research on the financial sustainability of high-speed railways.

**Keywords:** high-speed railway; financial sustainability; passenger flow sharing rate; model of the optimal ticket price

## 1. Introduction

The transportation industry an important service industry for the national economy; it allows people to pursue their livelihoods, and it is also the lifeline of the economy. As a result, it plays a key role in supporting and promoting the healthy and sustainable development of the national economy. Railways, especially high-speed railways have achieved rapid development in recent years and have become the beautiful "business card" in China's "The Belt and Road Initiative." Compared with aviation, high-speed railways have the advantages of a large transportation capacity and a convenient ride. Compared with highways and waterways, high-speed railways have the characteristics of high speed and little impact on the environment. Therefore, the influence of high-speed railways is self-evident. According to the "Medium-and Long-Term Railway Network Plan" issued by the National Development and Reform Commission, the Ministry of Transport, and the China Railway Corporation in July 2016, the scale of China's high-speed rail network would be expanded to "eight horizontal and eight vertical" main passages based on the original "four horizontal and four vertical" main passages. China also plans to build regional connecting lines of high-speed railways based on the "eight horizontal and eight vertical" main passages to further improve the transportation network and expand its coverage. The "National Integrated Three-Dimensional Transportation Network Planning Outline" (2021–2050) and other documents also set the standard for domestic

railway construction to enter the fast lane in the future. Judging from the state's investment in railway construction and according to the "Statistical Bulletin of the Chinese Railway Corporation" over the years, in 2018, investments in the fixed assets of the national railways were CNY 802.8 billion, and 4683 km of new lines were put into operation, including 4100 km of high-speed railways. In 2019, the investments were CNY 802.9 billion, and 8489 km of new lines were put into operation, including 5474 km of high-speed railways. In 2020, affected by the COVID-19, the investments decreased, but numbers still totaled CNY 781.9 billion, with 4933 km of new lines being put into operation, including 2521 km of high-speed railways. It can be seen that the state attaches great importance to railways, especially high-speed railways, in both policy planning and investment.

However, behind the continuous emergence of construction plans for high-speed railways, heavy debt pressure is a real problem facing China Railway Corporation. At present, construction funds for high-speed railways come mainly from two sources: one is equity capital, including special funds for railway construction, absorbed social capital, and local government investment; and the other is debt capital, such as bonds issued by China Railway Corporation and bank loans [1]. The latter requires a high return on investment, and the demand for debt funds is large, which has caused great pressure on enterprises to repay debts. According to the annual financial report, in 2020, China Railway Corporation had more than CNY 250 billion of non-current liabilities due within one year and more than CNY 1776.7 billion of bonds payable, while the net cash flow generated from operating activities was only about CNY 232.9 billion, and the net profit of that year was negative. Therefore, if we want to avoid repeating the tragedy that Taiwan faced when its high-speed railway was forced to restructure due to a serious financial crisis after less than three years of operation [2], it is very important for high-speed railway enterprises to enhance market competitiveness, improve profitability, ensure that the enterprises obtain a stable income and cash flow, and realize financial sustainability.

When analyzing the profitability of high-speed railways, the ticket price is one of the key indicators often studies by scholars. Zhao [3] proposed that whether the ticket price is scientific or not directly affects the profitability of high-speed railway enterprises, and we should further explore and extend price reform to achieve the highest returns for enterprises by scientifically formulating ticket prices. With the continuous advancement of corporation system reform of the railways, the right to set high-speed railway ticket pricing has been transferred from the government to the enterprises, greatly improving the operational autonomy of the enterprises. According to the current pricing standards for electric multiple unit (EMU) trains, the fares for a first-class seat and a second-class seat are determined by transportation enterprises based on relative laws and regulations, and the ticket price of a business class seat is determined independently according to market supply and demand and competition. On the one hand, railways have a certain responsibility for the public welfare, and the ticket price of a first-class seat and a second-class seat (which are public-oriented) cannot be inordinately high based merely the monopolistic characteristics of railways; there should be a "ceiling" on the price. On the other hand, high-speed railway enterprises are gradually being integrated into the transportation market and are allowed to participate in the competition, so it is impossible to ignore the market factors when setting prices. Thus, the ticket price should conform to the principle of maximizing the profits of enterprises to ensure that the enterprises can obtain stable and considerable profits. Therefore, how to make a reasonable ticket price is a question that needs to be answered when exploring the financial sustainability of high-speed railway transportation enterprises. Scholars have studied this issue from the perspectives of influencing factors, pricing methods, and policy measures. Regarding the issue of influencing factors, scholars have proposed that the competition with other modes of transportation (such as civil aviation), market demand and economies of scale, passenger choice, and other factors should be fully considered in the pricing of high-speed railway tickets. In a case study, Hu [4] established an optimized bi-level programming model by analyzing the competition between high-speed railways and civil aviation under the condition of marketization and

calculated the equilibrium ticket prices of high-speed railways under these competitive conditions. Lin [5] analyzed the impact of market demand and economies of scale on the pricing of high-speed railway tickets and proposed a pricing strategy for high-speed railways based on the principle of profit maximization. Jing et al. [6] fully considered the behavior of passenger choice to establish the pricing model for high-speed railways using the Beijing–Shanghai High-speed Railway as an example to calculate the model and obtained a reasonable ticket price. On the aspect of pricing methods, scholars generally referred to dynamic pricing methods. For example, Zhang et al. [7] established a Markov multidimensional decision-making model to make dynamic pricing decisions for high-speed railways. Song et al. [8] established a dynamic pricing model under the condition of the demand fluctuation that provides an optimized method for the dynamic pricing and ticket allocation for high-speed railways. On the aspect of policy measures, Yin [9] proposed that the relevant departments should relax the pre-price control and strengthen the post-price supervision against unfair competition and monopoly. However, these studies only focus on a certain aspect of pricing and fail to consider the important combination of factors that affect price, such as time value, supply and demand, competition, passenger utility, and national policies. Therefore, these research conclusions are often one-sided, cannot adequately reflect reality, and lack universality.

Thus, this paper constructs a model of passenger flow sharing rate that reflects supply and demand, market competition, and passenger utility, and applies it to the model of the optimal ticket price for high-speed railways. Additionally, the model uses the listed Beijing–Shanghai High-speed Railway as an example for calculating the optimal ticket price, thus providing a method with a certain reference value for the pricing of high-speed railways. Moreover, using the optimal ticket price calculated, this paper quantitatively analyzes the impact of ticket price changes on the financial sustainability of enterprises and explores the path for high-speed railway transportation enterprises to achieve financial sustainability.

The main contributions of this study are as follows: This paper (1) constructs a model of passenger flow sharing rates for high-speed railways and integrates it into the optimal pricing model so that the ticket price can better reflect market information; (2) reveals the relationship between ticket price and financial sustainability and constructs an framework for the analysis of financial sustainability centered on the ticket price.

In the second part of this paper, the literature review is carried out. In the third part, the impact of the mechanism of the ticket price on financial sustainability is expounded. In the fourth part, taking the Beijing–Shanghai High-speed Railway as an example, the paper calculates the pricing model, obtains the optimal ticket price, and analyzes its impact on the financial sustainability of the enterprises. Finally, the paper summarizes the research conclusions and points out the shortcomings.

## 2. Literature Review

The construction and opening of a high-speed railway would not only make travel more convenient, but would also play a positive role in promoting local economic development and environmental improvement. Fan [10] found that the opening of high-speed railways has significantly curbed smog pollution; Sun [11] stated that high-speed railways not only promote urban economic development but also reduce urban pollutant emissions. China's central and local governments have a very positive attitude towards the construction of high-speed railways, and the investment in these projects has increased annually [12]. In fact, the amount of funds needed for the construction of high-speed railways is huge. However, the special funds available for railways and the financial input from the central and local governments are very limited. Therefore, it is necessary to absorb social capital with a high return on investment and secure bank loans with high-interest rates, and the resulting pressure of repaying principal and interest faced by China Railway Corporation is greater every year. If stable income cannot be guaranteed and cash flow is cut off, it will have an adverse impact on the sustainable development of enterprises and high-speed railways. Some scholars have analyzed the profitability and financing

mode of high-speed railways from the perspective of economics and pointed out that most high-speed railways are in a state of financial loss at present; some of them cannot even achieve profit within the initial ten years of operation [13]. Therefore, most high-speed railways in China are debt-oriented, and the current ticket prices and traffic volumes are obviously not conducive to the balanced financial development of high-speed railways, so most enterprises face the dilemma of great debt repayment pressure in the early stages of development and great operating pressure in the later stages [14]. An indicator system should be constructed for evaluating the financial sustainability of high-speed railways that can comprehensively analyze the ticket price, operating income, cost, and traffic volume from the perspectives of debt repayment, daily operation, and future development, and accurately evaluate the profitability and development potential of high-speed railways [15]. At present, most scholars carry out financial analysis and evaluation on infrastructure construction and operation for PPP projects. Gao [16] found that infrastructure construction projects carried out by the PPP mode should set up an indicator system for financial evaluation from four aspects: profitability, return on investment, financial risk, and solvency. Similarly, Huang [17] constructed a financial evaluation model of PPP projects and used the analytic hierarchy process to evaluate the project financially by taking these four dimensions as the first-level indicators, with the investment profit rate, the internal rate of return, the accumulated cash flow, and the asset-liability ratio as the second-level indicators. Wang [18] compared the differences between traditional rail transit projects and PPP projects from the aspects of the financial evaluation of purpose and scope, the income distribution mode, the analysis system, and the freight rate-setting method. It is proposed that the financial evaluation system of PPP projects should focus on three aspects: the preliminary analysis of the financial evaluation, the calculation of the feasibility gap, and the analysis of the financial affordability of the projects. Zhou and Liu [19] found that in the financial and economic evaluation of infrastructure projects, it was necessary to focus on the analysis of financing and risks in order to comprehensively examine the profitability and economic rationality.

In conclusion, on the one hand, although the existing financial evaluation research on infrastructure construction has been rich, the research on the financial sustainability of high-speed railways is still relatively sparse, and most of the studies are only qualitative research, lacking quantitative analysis. On the other hand, scholars mainly discuss the subject of building a system of financial evaluation for the infrastructure, but few people focus on a specific factor that affects the financial situation and analyze its relevance to the financial sustainability of high-speed railways layer by layer. In fact, the high-speed railway transportation costs, market competition, and passenger's travel preferences affect the ticket price, and the ticket price affects the passenger flow; both of these are closely related to the profitability of the enterprises, and the ticket income is the main source of operating income for high-speed railways. To explore the sustainable development path of enterprise finance and improve profitability, the passenger flow sharing rate and the ticket price of high-speed railways are very important. Compared with domestic enterprises, foreign high-speed railways appeared earlier, and the research on the sharing rate and pricing is relatively mature. Gonzalez-Savignat [20] and Aoife et al. [21], respectively, studied the Barcelona–Madrid high-speed railway in Spain and the Irish passenger transport corridor and concluded that the running time and ticket price of high-speed railways are their competitive advantages and the reason for the large sharing rate of high-speed railways. Dobruszkes and Givoni [22] suggested that in addition to the running time, the high load rate of high-speed railways is also a factor that makes them superior to civil aviation. Agostino. N et al. [23] established a choice model of the passenger's traveling mode, simulated the influence of the change of service attributes of railway transport on a passenger's choice of the service type and class, and studied the policy of optimal ticket price. Ozbay et al. [24] calculated the corresponding travel time value by train arrival time and the passenger's travel purposes and then used the pricing model to calculate the corresponding ticket price. Lin [25] and Crevier [26] put forward relevant suggestions on

the marketing strategy and price management of railways after they studied their pricing methods and mechanisms.

In addition to studying the passenger flow sharing rate and impact factors on the pricing of high-speed railways, Chinese scholars also focused on the research concerning the pricing methods of high-speed railways. Although two high-speed railway corporations, Beijing–Shanghai and Guangzhou-Shenzhen, have been listed in China, the current ticket pricing methods generally lack the embodiment of competitive factors. Yang and Zhang [27] found that the basis for formulating the ticket prices of high-speed railways in China is relatively simple, considering only distance factors without including market demand information; thus, they cannot reflect the relationship between supply and demand in the transportation market. Therefore, the formulation of ticket prices should be based on the theory of revenue management, adhere to the principle of maximizing revenue, pay attention to the transportation market, target customer groups, services, and other factors, evaluate foreign pricing methods, and implement flexible pricing strategies. Chen and Lin [28] fully analyzed the ticket pricing mechanism of high-speed railways and civil aviation and proposed that high-speed railways should determine ticket prices using a competitive pricing method so that the price can better reflect competitiveness. Yu [29] put forward the dynamic pricing strategy for high-speed railways by establishing a dynamic expectation model. In addition to the dynamic pricing method, the differential pricing method [30,31], which includes factors such as the passenger's travel purpose and psychological demands, and the market-adjusted pricing method [32], which can reflect fairness and efficiency, are also commonly mentioned by domestic scholars.

In conclusion, although many scholars have carefully studied the importance of the profitability of infrastructure projects such as high-speed railways, as well as ways to improve profitability and the construction of financial evaluation systems, putting forward numerous targeted policy recommendations for realizing the scientific financial evaluation and sustainable development of high-speed railway transportation enterprises, most of the current research remains at the stage of describing the present situation using qualitative analysis. Quantitative research, especially research considering financial sustainability from the perspective of pricing is relatively limited. Moreover, although scholars at home and abroad have carried out abundant research on the ticket pricing of high-speed railways, there are few studies investigating pricing methods that integrate market competition, transportation cost and benefit, passenger's travel choice preferences, and other factors that affect pricing into a pricing model. Therefore, taking into account the factors such as competition, passenger preference, and so on, this paper constructs the model of the passenger flow sharing rate and the optimal pricing model of high-speed railways based on the principle of maximizing the profits of the enterprises. The Beijing–Shanghai High-speed Railway, a representative high-speed railway connecting Beijing and Shanghai and passing through Tianjin, Jinan, Nanjing, Suzhou, and other first-tier cities, is used as the research object for calculating the optimal pricing model. Using the calculated optimal ticket price, this paper analyses the impact of the ticket price on financial sustainability and explores the means to realize the financial sustainability of high-speed railway transportation enterprises.

## 3. The Effect of the Mechanism of Ticket Price on the Financial Sustainability, the Optimal Pricing Model, and the Construction of an Evaluation System of Financial Sustainability

### 3.1. The Effect of the Mechanism of Ticket Price on Financial Sustainability

The word "sustainable" means that a thing retains a state in time and space for a long time without interruption. The financial sustainability of high-speed railways is the process that the cash flow generated by production and operation activities can maintain normal operation, meet the current repayment of principal and interest, and meet the requirements of the return on investment in the whole life cycle of the enterprise [33]. For high-speed railroad transportation enterprises, ticket price and passenger flow determine the operating income of enterprises, and the operating income is an important indicator reflecting the "hematopoiesis" function of the cash flow of enterprises, so the ticket price affects the

financially sustainable development of enterprises. The research [34] found that there is a positive relationship between the price and the sustainable growth rate of the company, and the fluctuation of price has an impact on financial indicators, such as the net profit margin on sales, when studying the impact of crude oil price on the sustainable growth rate of the China National Petroleum Corporation. As a result, the ticket prices of high-speed railways should also have an impact on the financial sustainability of enterprises.

Although the ticket pricing of high-speed railways adopts the method of "flexible pricing" at present, the price is determined from the perspective of providing better service for passengers according to the market supply and demand relationship [35]. From an actual implementation standpoint, it is not ideal. In fact, the current high-speed railway ticket prices cannot fully reflect the market demand and competition, and the lack of a flexible price-adjusted mechanism makes it impossible for transportation enterprises to make timely adjustments according to market changes. This affects not only passenger flow, but also the realization of income [36]. Therefore, it is necessary to construct a pricing model that can reflect the market information and passenger service and also guarantee the operating efficiency of high-speed railroad transportation enterprises. The relationship between the ticket price and financial sustainability, and the concept of creating an optimal pricing model is shown in Figure 1.

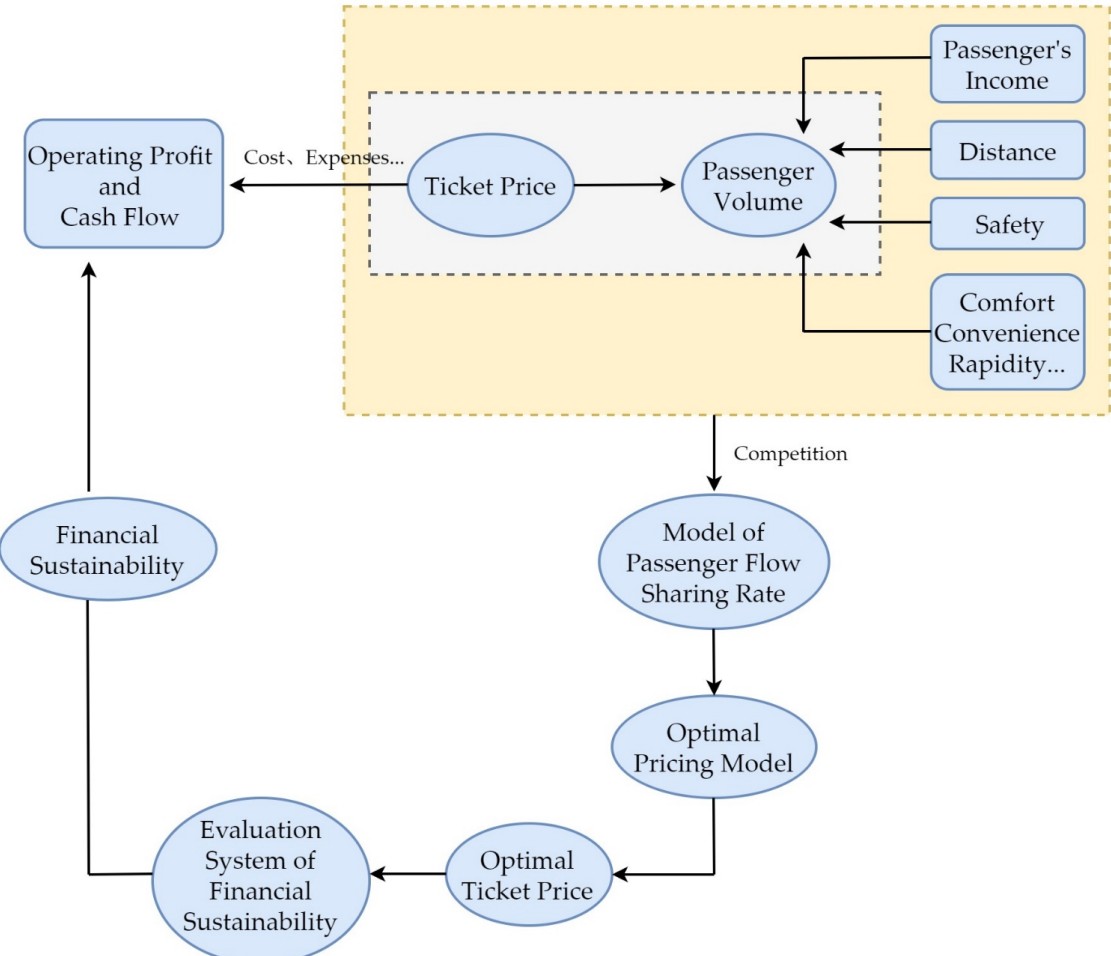

**Figure 1.** The Analysis Framework of the Relationship between Ticket Price and Financial Sustainability and the Construction of an Optimal Pricing Model.

Figure 1 shows that the ticket price has an effect on financial sustainability by affecting the operating profit and cash flow of the enterprises. We should construct a pricing model based on the principle of profit maximization on the basis of fully considering market

supply and demand, competition, and a passenger's traveling choice so that we can obtain an optimal ticket price and explore the impact of this price on financial sustainability using the constructed evaluation system of financial sustainability.

### 3.2. The Construction of an Optimal Pricing Model

3.2.1. Model of Passenger Flow Sharing Rate

For transportation enterprises, the passenger flow sharing rate is a reflection of market competitiveness. Factors that affect the passenger flow sharing rate can be analyzed from two aspects: demand and supply. From the perspective of demand, the passenger's income, the price sensitivity, and the consideration of comfort and safety are factors that belong to the passenger's characteristics that affect the sharing rate; the purpose, distance, and time of the passenger's travel are also factors related to the passenger's traveling characteristics. From the perspective of supply, the safety, economy, speed, comfort, and convenience of various modes of transportation are factors related to the technical and economic characteristics of these modes of transportation that affect the sharing rate [37]. This paper constructs a function of observable cost to quantify the above-mentioned interfering factors first and then, based on the economic hypothesis of rational man, passengers choose different modes of transportation to maximize utility. Drawing lessons from Kong [38] and other scholars' research findings for calculating the passenger flow sharing rate, this paper constructs the logit model as Equation (1) shows:

$$P_i = \frac{\exp(-R_i)}{\sum_{i=1}^{n} \exp(-R_i)} \tag{1}$$

$P_i$ represents the passenger flow sharing rate of the $i$th mode of transportation; $R_i$ represents the observable cost of the $i$th mode of transportation; $n$ represents the number of types of transportation options available for passengers.

In order to eliminate the expansion effect of exponential growth on the results, this paper will average Equation (1) as follows:

$$P_i = \frac{\exp(-R_i/\overline{R})}{\sum_{i=1}^{n} \exp(-R_i/\overline{R})} \tag{2}$$

$\overline{R}$ represents the average observable cost, which can be calculated as follows:

$$\overline{R} = \frac{\sum_{i=1}^{n} R_i}{n} \tag{3}$$

According to the previous analysis, the factors that affect the passenger flow sharing rate of various transportation modes can be summarized as safety, economy, speed, convenience, and comfort. Drawing lessons from the research findings of Li [39], this paper brings these five factors into the function of observable cost, as is shown in Equation (4):

$$R^i{}_{AB} = \frac{1}{S_{i1}} \times (\mu_1 S_{i2} + \mu_2 S_{i3} + \mu_3 S_{i4} + \mu_4 S_{i5}) \tag{4}$$

$R^i{}_{AB}$ represents the observable cost of the $i$th mode of transportation from place $A$ to place $B$; $\frac{1}{S_{i1}}$ represents the observable cost of safety; $S_{i2}$, $S_{i3}$, $S_{i4}$, and $S_{i5}$ are the observable cost of economy, rapidity, convenience, and comfort respectively; $\mu_1$, $\mu_2$, $\mu_3$, and $\mu_4$ is the weight of each observable cost respectively.

Since safety is the most basic factor considered by passengers when choosing a mode of travel, the impact on utility or cost is independent. Therefore, according to the principle of multiplication, in the function of observable cost, the cost of safety is multiplied in the function to reflect its overall impact on the observable cost. $S_{i1}$ indicates the degree to which passengers expect their personal and property safety to be protected when they choose a certain mode of transportation [40] with the numerical value ranges from 0 to 1. The larger

the numerical value, the higher the security, and the smaller $\frac{1}{S_{i1}}$, the more likely passengers will be willing to choose this mode of transportation.

The observable cost of economy, $S_{i2}$, represents the cost that passengers are willing to pay for choosing a certain mode of transportation, that is, the freight rate. For the convenience of calculation, this paper uses the ticket price, $p_{i2}$, to express $S_{i2}$.

The observable cost of rapidity, $S_{i3}$, can be measured by the total traveling time of passengers. "Total traveling time" includes the time from the place of departure to the initial station, the running time of the transport vehicles, and the time from the final station to the destination, which can be expressed by Equation (5).

$$S_{i3} = \left( t_{i3}{}^0 + t_{i3}{}^1 + t_{i3}{}^2 \right) \times vot \tag{5}$$

$t_{i3}{}^0$, $t_{i3}{}^1$, and $t_{i3}{}^2$ are the time from the place of departure to the initial station, the running time of the transport vehicles, and the time from the final station to the destination, respectively; $vot$ represents the time value of the passenger's travel. This paper uses the model discussed by Ding [41] to estimate the time value, which can be expressed by Equation (6).

$$vot = \frac{GDP}{T \times PN} \tag{6}$$

*GDP*, *PN*, and *T* represent the gross domestic product, the population, and the average working hours of the workers of a certain area.

Since traveling is bidirectional and involves at least two cities, origin *A* and destination *B*, in order to facilitate statistics, this paper assumes that the average passenger volume between *A* and *B* is equal. Then, the time value of passengers from place *A* to place *B* can be expressed by the average value of the two places. The calculation formula is shown in Equation (7).

$$vot_{AB} = (vot_A + vot_B)/2 \tag{7}$$

The convenience reflects the complexity of the passenger's boarding or check-in procedures, as well as the length of time to wait for the arrival of the means of transportation. The degree of convenience can be measured by the time spent [38]. Based on the research about the measurement of convenience proposed by Zhang [37], the observable cost of convenience, $S_{i4}$, can be represented by the time for passengers to check in and wait for boarding, which is similar to the quantitative method of $S_{i3}$. This paper uses $t_{i4}{}^4$ and $t_{i4}{}^5$ to represent these two parts of time. $S_{i4}$ can be estimated by Equation (8).

$$S_{i4} = \left( t_{i4}{}^4 + t_{i4}{}^5 \right) \times vot_{AB} \tag{8}$$

The observable cost of comfort, $S_{i5}$, is inversely proportional to the time required for passengers to eliminate fatigue, that is, the longer the time for passengers to eliminate fatigue, the worse the comfort. Learning from the common practice of most scholars such as Zhang [37] and Kong [38], $S_{i5}$ can be estimated by Equation (9).

$$S_{i5} = \frac{H}{1 + \alpha_i \times e^{-\beta_i \times t_{AB}{}^i}} \times vot_{AB} \tag{9}$$

$\frac{H}{1 + \alpha_i \times e^{-\beta_i \times t_{AB}{}^i}}$ represents the time required for passengers to eliminate fatigue. Based on the general psychology of human beings, people's sensitivity to fatigue will decrease over time, and the change in sensitivity is non-linear. Thus, this paper uses $\frac{1}{1 + \alpha_i \times e^{-\beta_i \times t_{AB}{}^i}}$ to reflect this general psychological phenomenon. And *H* is the limit time for passengers to eliminate fatigue. According to the thoughts of Kong [38], the time for people to eliminate fatigue will not extend indefinitely. Therefore, *H* is a constant. $\alpha_i$ and $\beta_i$ are undetermined coefficients. When $t_{AB}{}^i$ is equal to zero, the minimum time for passengers to eliminate fatigue is equal to $\frac{H}{1 + \alpha_i}$; $\beta_i$ represents the degree of fatigue recovery per unit time, the

bigger the $\beta_i$ is, the longer the time for passengers to eliminate fatigue [42]. $t_{AB}{}^i$ is the sum of $t_{i3}{}^0$, $t_{i3}{}^1$, $t_{i3}{}^2$, $t_{i4}{}^4$ and $t_{i4}{}^5$, which equals the total traveling time for passengers from $A$ to $B$.

As a result, $R^i{}_{AB}$, the observable cost of the $i$th mode of transportation selected by passengers from $A$ to $B$ is quantified, and by putting $R^i{}_{AB}$ into Equation (2), the passenger flow sharing rate of various modes of transportation can be obtained.

### 3.2.2. The Construction of an Optimal Pricing Model

As mentioned below, the ticket price affects the passenger flow sharing rate, which ultimately influences the operating income and benefits of the transportation enterprises. Wang [43] studied the ticket prices of buses and found that when the ticket price is reduced by 37.5%, the passenger volume is increased by 22.5%; the social benefits are increased as well, but the total benefits of the operators are reduced. Zhang [44] analyzed the relationships among the ticket price of high-speed railways, the passenger flow sharing rate, and the operating income, and estimated the ticket price, distinguishing between peak and non-peak periods. It can be seen that there is a close relationship between the ticket price, the passenger flow sharing rate, and the operating income. At present, some scholars use the two-level programming model to study the pricing of high-speed railways [45,46]. Although this model fully considers the principle of maximizing revenue, it cannot reflect the linkage relationship between the ticket price and the passenger flow sharing rate under market competition. Therefore, the method of constructing a pricing model for high-speed railways in this paper is as follows: Firstly, we express the changes in passenger volume as a function of the observable cost and the passenger flow sharing rate and combine the price to construct the function of the operating income; secondly, according to the principle of profit maximization, the goal is to "maximize the numerical value of operating income minus operating cost" to form an optimal pricing model for high-speed railways.

- The Model of Passenger Volume

Generally speaking, the increase in passenger volume of a mode of transportation can be roughly attributed to two aspects. One is due to technical changes, which can make it faster and more comfortable than other modes of transportation; the other is due to the opening of new lines, which can attract passengers by providing them with more modes of transportation to choose from. The former is called induced passenger volume, and the latter is called transferred passenger volume. Based on the research of Kong [38] and Wang [47], the induced passenger volume $Q_{AB(INDUCE)}{}^i$ can be expressed by Equation (10) as follows:

$$Q_{AB(INDUCE)}{}^i = k \times \frac{(G_A \times PN_A)^\alpha \times (G_B \times PN_B)^\beta}{(R_{AB}{}^i)^\gamma} \tag{10}$$

$k$, $\alpha$, $\beta$, and $\gamma$ are parameters; $G_A$ and $G_B$ are the *GDP* of place $A$ and place $B$; $PN_A$ and $PN_B$ are the population of place $A$ and place $B$.

For induced passenger volume, the GDP and population of place A and place B will not cause changes. The change in ticket price affects the passenger flow sharing rate, making the observable cost change, thus affecting the induced passenger volume. Therefore, when the observable cost $R_{AB}{}^i$ changes, the rate of change of the induced passenger volume can be expressed by Equation (11).

$$r_{INDUCE}{}^i = \left[ \frac{R_{AB}{}^i}{(R_{AB}{}^i)'} \right]^\gamma - 1 \tag{11}$$

$r_{INDUCE}{}^i$ is the rate of change of the induced passenger volume between place $A$ and place $B$; $(R_{AB}{}^i)'$ is the change in the observable cost due to the change of the ticket price; $\gamma$ is the undetermined coefficient, which is generally taken as 0.709 [48].

Drawing lessons from the research of Kong [38] and Wang [49], the transferred passenger volume can be expressed by Equation (12).

$$Q_{AB(TRANSFER)}{}^i = Q_{AB(TOTAL)} \times P_i = Q_{AB(TOTAL)} \times \frac{\exp\left(-R_i/\overline{R}\right)}{\sum_{i=1}^n \exp\left(-R_i/\overline{R}\right)} \tag{12}$$

$Q_{AB(TOTAL)}$ is the total passenger volume between $A$ and $B$; $P_i$ is the sharing rate of transferred passenger volume.

Similar to the induced passenger volume, the rate of change of the transferred passenger volume can be expressed by Equation (13).

$$r_{TRANSFER}{}^i = \frac{P_i'}{P_i} - 1 = \frac{\dfrac{\exp\left(-\frac{R_i'}{\overline{R}'}\right)}{\sum_{i=1}^n \exp\left(-\frac{R_i'}{\overline{R}'}\right)}}{\dfrac{\exp\left(-\frac{R_i}{\overline{R}}\right)}{\sum_{i=1}^n \exp\left(-\frac{R_i}{\overline{R}}\right)}} - 1 \tag{13}$$

$r_{TRANSFER}{}^i$ is the rate of change of the transferred passenger volume between place $A$ and place $B$; $R_i'$ and $\overline{R}'$ are the observable cost and average observable cost that have changed due to the change of ticket price.

- The Optimal Pricing Model

After obtaining the passenger flow sharing rate and the rate of change of the passenger volume, the profit function of the transportation enterprises can be expressed by Equation (14).

$$\pi = p_i' \times q_i \times \left(1 + r_{INDUCE}{}^i + r_{TRANSFER}{}^i\right) - c_i \tag{14}$$

$\pi$ represents the profits of the enterprises; $p_i'$ is the ticket price to be estimated in this paper; $q_i$ is the passenger volume before the ticket price and the passenger flow sharing rate changed; $c_i$ is the operating cost. Incorporating Equations (11) and (13) into Equation (14), we can get the profit function of the $i$th mode of transportation from place $A$ to place $B$:

$$d\pi = p_i' \times q_i \times \left[\left(\frac{R_i}{R_i'}\right)^\gamma + \frac{\dfrac{\exp\left(-\frac{R_i'}{\overline{R}'}\right)}{\sum_{i=1}^n \exp\left(-\frac{R_i'}{\overline{R}'}\right)}}{\dfrac{\exp\left(-\frac{R_i}{\overline{R}}\right)}{\sum_{i=1}^n \exp\left(-\frac{R_i}{\overline{R}}\right)}} - 1\right] - c_i \tag{15}$$

Therefore, from the perspective of the enterprises, with the goal of maximizing profits [38,50], the optimal pricing model of the $i$th mode of transportation is:

$$\max(\pi) = p_i' \times q_i \times \left[\left(\frac{R_i}{R_i'}\right)^\gamma + \frac{\dfrac{\exp\left(-\frac{R_i'}{\overline{R}'}\right)}{\sum_{i=1}^n \exp\left(-\frac{R_i'}{\overline{R}'}\right)}}{\dfrac{\exp\left(-\frac{R_i}{\overline{R}}\right)}{\sum_{i=1}^n \exp\left(-\frac{R_i}{\overline{R}}\right)}} - 1\right] - c_i \tag{16}$$

$$\text{s.t.} \, p_i' \gg p_{min} \tag{17}$$

$$I' \gg I \tag{18}$$

Equations (17) and (18) are the constraints; $p_{min}$ is the minimum numerical value of the floating price of the $i$th mode of transportation in the past years; $I$ is the operating income gained under the current ticket price; $I'$ is the operating income gained under the optimal ticket price.

### 3.3. The Construction of an Evaluation System of Financial Sustainability

In the previous review of relevant literature, it is not difficult to find that scholars' discussions on the financial sustainability of high-speed railways are mostly focused on "what affects the financial sustainability of high-speed railways," and "how to achieve the financial sustainability of high-speed railways." However, few scholars have done systematic research on the evaluation of financial sustainability from the perspective of the ticket price. In order for us to comprehensively explore the path to achieve the financial sustainability of high-speed railways, Ding [15] proposed that the evaluation system of financial sustainability of high-speed railways should be established from three aspects: sustainable operation, sustainable debt repayment, and sustainable development. This paper draws on his ideas to establish an evaluation system of financial sustainability of high-speed railways that takes the ticket price as a center. The evaluation system is shown in Table 1.

**Table 1.** The Evaluation System of the Financial Sustainability of High-speed Railways [1].

| | Index | Formula |
|---|---|---|
| Sustainable Operation | Gross Margin | $\dfrac{operating\ income - operating\ cost}{operating\ income}$ |
| | Operating Profit Margin | $\dfrac{operating\ income - operating\ cost - G\ and\ A\ expense}{operating\ income}$ |
| Sustainable Debt Repayment | Ratio of Debt Repayment | $\dfrac{Net\ Cash\ Flow\ from\ Operating}{Total\ Liabilities}$ |
| | Current Ratio | $\dfrac{Net\ Cash\ Flow\ from\ Operating}{Current\ Liabilities}$ |
| Sustainable Development | Growth Rate of Operating Income | $\dfrac{operating\ income\ under\ p_i' - operating\ income\ under\ p_i}{operating\ income\ under\ p_i}$ |

[1] In this table, the indexes and definitions of formulas come from Ding [15].

In the table, the sustainable operation examines the ability of high-speed railroad transportation enterprises to obtain stable income while conducting daily operating activities and maintaining normal operations. The gross margin and the operating profit margin can reflect the true profitability of an enterprise and the level of performance that can be improved by adjusting the price, which is an intuitive manifestation of the sustainability of operations. Therefore, this paper chooses these two indicators to reflect the changes in the sustainability of operations before and after the ticket price change.

Sustainable debt repayment examines the ability of high-speed railroad transportation enterprises to repay their debts on time. The ratio of debt repayment and the current ratio are indicators that reflect the ability of enterprises to repay short-term and long-term debt and reveal the level of liquidity risk. The higher the two ratios, the better the liquidity of assets and the stronger the solvency. This paper selects these two indicators to fully reflect the changes in debt solvency caused by the changing of the ticket price from both short-term and long-term perspectives.

Sustainable development examines the ability of high-speed railroad transportation enterprises to develop and grow on a healthy and long-term basis. Whether sustainable development can be achieved or not, in the final analysis, depends on the company's operating conditions and profitability. Therefore, this paper chooses the growth rate of operating income as an indicator to reflect the sustainability of development.

## 4. Case Analysis on Beijing–Shanghai High-Speed Railway

At present, China has two listed high-speed railway companies, the Beijing–Shanghai High-speed Railway and the Guangzhou-Shenzhen Railway; both of these are mainly engaged in passenger transportation. In addition to providing passenger transportation services, the business scope of Guangzhou-Shenzhen Railway also includes cargo transportation. However, the main business of the Beijing–Shanghai High-speed Railway is high-speed passenger transportation that does not involve normal-speed cargo and passenger transportation. Therefore, this paper takes Beijing–Shanghai High-speed Railway as the research object to calculate the optimal ticket price under the conditions of market competition.

The Beijing–Shanghai High-speed Railway starts in Beijing, passes through Tianjin, Jinan, Bengbu, Nanjing, Wuxi, Suzhou, and other first-tier cities, and ends at Shanghai. As far as the route from Beijing to Shanghai is concerned, in addition to high-speed railways, there are also three other modes of transportation for passengers to choose from: civil aviation, normal-speed rail, and highway. However, at this stage, there is only one bus service and two normal-speed railway departures from Beijing to Shanghai every day. Compared with the high-speed railway and civil aviation options, which have many schedules and flights, the market share of the normal-speed railway and highway is negligible. Therefore, this paper only considers the competition between high-speed railways and civil aviation. To facilitate the discussion below, this paper stipulates that in the aforementioned model, when $i$ is 1, it means high-speed railway; when $i$ is 2, it means civil aviation.

In addition, due to the impact of COVID-19, the transportation industry was seriously hit in 2020. In that year, the number of trips and people's willingness to travel were significantly reduced; therefore, the arrangements of high-speed railway schedules and civilian flights were reduced to varying degrees. In order to make the research conclusions more general, the public data used in the case analysis of this paper are all selected from relevant statistics from 2019.

*4.1. The Calculation of the Passenger Flow Sharing Rate of the Beijing–Shanghai High-Speed Railway*

4.1.1. The Selection of Research Objects

There are twenty-three cities along the Beijing–Shanghai High-speed Railway, including Beijing, Langfang, Tianjin, Cangzhou, Dezhou, Jinan, Tai'an, Qufu, Tengzhou, Zaozhuang, Xuzhou, Suzhou, Bengbu, Dingyuan, Chuzhou, Nanjing, Zhenjiang, Danyang, Changzhou, Wuxi, Suzhou, Kunshan, and Shanghai. Among these, ten cities have civil airports, namely Beijing, Tianjin, Jinan, Qufu, Xuzhou, Bengbu, Nanjing, Changzhou, Wuxi, and Shanghai. Since this paper focuses mainly on the competition between high-speed railways and civil aviation and considering that non-direct air routes are less convenient than direct routes, this paper selects seven cities (from the ten listed) with direct flights as the research objects to calculate the passenger flow sharing rate and the optimal ticket price for the Beijing–Shanghai High-speed Railway. The direct flights among the seven cities are shown in Table 2.

**Table 2.** Statistics of Direct Flights between the Seven Cities [1,2].

| City | Nanjing | Changzhou | Wuxi | Shanghai |
|---|---|---|---|---|
| Beijing | Y | Y | Y | Y |
| Tianjin | N | N | N | Y |
| Jinan | N | N | N | Y |

[1] *Y* indicates that there are direct flights between the two cities, and *N* indicates that there are no direct flights between the two cities. [2] Source: https://www.ctrip.com/ (accessed on 18 December 2021).

4.1.2. The Calculation of the Observable Cost

As mentioned above, the factors that affect the passenger flow sharing rate can be summarized as safety, economy, rapidity, convenience, and comfort. Correspondingly, the function of the observable cost includes five parameters; the calculation of each parameter is shown as follows.

- The Observable Cost of Safety $\frac{1}{S_{i1}}$

Liu [51] has studied the measuring method of the risk rate of various modes of transportation and proposed that the risk rates of high-speed railways and civil aviation are 0.003 percent and 0.0046 percent, respectively. The two numerical values are very small and close, which can be regarded that as for the safety degree of the Beijing–Shanghai High-speed Railway, $S_{11}$, is almost the same as that for civil aviation, $S_{21}$, and that the numerical

value is close to 1. Therefore, the observable cost of safety, $\frac{1}{S_{11}}$ for the Beijing–Shanghai High-speed Railway, and $\frac{1}{S_{21}}$ for civil aviation, are approximately equal to 1.

- The Observable Cost of Economy $S_{i2}$

This paper uses the ticket price, $p_{i2}$, to measure the observable cost of the economy. However, EMU trains have three types of seats: first-class, second-class, and business class. Civil aviation also has three classifications of seats: economy class, business class, and first-class. Fares for different classes of seats vary. This paper only takes the ticket price of a second-class seat on the Beijing–Shanghai High-speed Railway, $p_{12}$, and the fare for an economy class civil aviation, $p_{22}$, to measure the observable cost of economy, $S_{12}$ and $S_{22}$, respectively, and the calculated results are shown in Tables 3 and 4.

**Table 3.** The Observable Cost of Economy for the Beijing–Shanghai High-speed Railway $S_{12}$ (unit: CNY) [1].

| City | Nanjing | Changzhou | Wuxi | Shanghai |
|---|---|---|---|---|
| Beijing | 464 | 513 | 535 | 631 |
| Tianjin | 423 | 478 | 488 | 541 |
| Jinan | 278 | 345 | 370 | 417 |

[1] Source: https://www.ctrip.com/ (accessed on 18 December 2021).

**Table 4.** The Observable Cost of Economy for Civil Aviation $S_{22}$ (unit: CNY) [1].

| City | Nanjing | Changzhou | Wuxi | Shanghai |
|---|---|---|---|---|
| Beijing | 780 | 795 | 815 | 870 |
| Tianjin | N | N | N | 850 |
| Jinan | N | N | N | 690 |

[1] Source: https://flight.qunar.com/ (accessed on 18 December 2021).

- The Observable Cost of Rapidity $S_{i3}$

Table 3. first of all, it is necessary to estimate the time from the place of departure to the initial station, $t_{i3}^0$, the running time of the transport vehicles, $t_{i3}^1$, and the time from the final station to the destination, $t_{i3}^2$. There is no publicly available data for $t_{i3}^0$ and $t_{i3}^2$, and for different passengers, the income, occupation, and travel purpose will cause differences in $t_{i3}^0$ and $t_{i3}^2$. In order to simplify the calculation, this article draws on the measuring method proposed by Zhang [37] to set $t_{i3}^0$ equal to $t_{i3}^2$, obtaining the total transition time $(t_{i3}^0 + t_{i3}^2)$ through Zhang's investigation and other correlating data. The results are shown in Tables 5–7.

**Table 5.** The Total Transition Time $(t_{i3}^0 + t_{i3}^2)$ (unit: hour).

| Total Transition Time | High-Speed Railway $(t_{13}^0+t_{13}^2)$ | Civil Aviation $(t_{23}^0+t_{23}^2)$ |
|---|---|---|
| $t_{i3}^0 + t_{i3}^2$ | 2.02 | 2.54 |

**Table 6.** The Running Time of the Beijing–Shanghai High-speed Railway $t_{13}^1$ (unit: hour) [1].

| City | Nanjing | Changzhou | Wuxi | Shanghai |
|---|---|---|---|---|
| Beijing | 4.4 | 5.08 | 5.25 | 6.1 |
| Tianjin | 4.06 | 4.75 | 5.08 | 5.85 |
| Jinan | 2.91 | 3.3 | 3.7 | 4.3 |

[1] Source: https://www.12306.cn/ (accessed on 18 December 2021).

**Table 7.** The Running Time of Civil Aviation $t_{23}$[1] (unit: hour) [1].

| City | Nanjing | Changzhou | Wuxi | Shanghai |
|---|---|---|---|---|
| Beijing | 1.92 | 2 | 2.16 | 2.41 |
| Tianjin | N | N | N | 2.08 |
| Jinan | N | N | N | 1.67 |

[1] Source: https://flight.qunar.com/ (accessed on 18 December 2021).

To calculate the time value, *vot*, we need to obtain the *GDP* of each city, the regional population number, *PN,* and the average labor time, *T,* of the laborers. Relevant 2019 socio-economic data from each city is shown in Table 8.

**Table 8.** The GDP and the Population of Each City in 2019 (unit: hundred million CNY; ten thousand people) [1].

| City | Beijing | Tianjin | Jinan | Nanjing | Changzhou | Wuxi | Shanghai |
|---|---|---|---|---|---|---|---|
| GDP | 35,371.3 | 14,104.28 | 9443.4 | 14,030.15 | 7400.9 | 11,852.32 | 38,155.32 |
| Population (PN) | 2153.5 | 1556.87 | 846.62 | 880.67 | 548.8 | 683.05 | 2418.34 |

[1] Source: http://www.sohu.com/; http://www.phb123.com/ (accessed on 18 December 2021).

According to the provisions of the "Decision of the State Council on Revising the National Holidays and Memorial Day, the Third Revision" 11 December 2013, the average monthly working days and average monthly working hours for Chinese employees in 2019 were 20.83 days and 166.64 h. Therefore, in 2019, the average working hours for employees in China was 1999.68 h. Thus, we can calculate the time value of passengers in each city, as shown in Table 9.

**Table 9.** The Time Value of Passengers in Each City (unit: CNY/(person × hour)).

| City | Beijing | Tianjin | Jinan | Nanjing | Changzhou | Wuxi | Shanghai |
|---|---|---|---|---|---|---|---|
| vot | 82.14 | 45.3 | 55.78 | 79.67 | 67.44 | 86.77 | 78.9 |

Based on this, according to Equation (7), the time value of passengers among cities, $vot_{AB}$, is shown in Table 10.

**Table 10.** The Time Value of Passengers among Cities $vot_{AB}$ (unit: CNY/(person × hour)).

| City | Nanjing | Changzhou | Wuxi | Shanghai |
|---|---|---|---|---|
| Beijing | 80.9 | 74.79 | 84.46 | 80.52 |
| Tianjin | 62.49 | 56.37 | 66.04 | 62.1 |
| Jinan | 67.72 | 61.61 | 71.28 | 67.34 |

Combining the results of Tables 5–7 and Table 10, the observable cost of rapidity, $S_{i3}$, can be shown in Tables 11 and 12.

**Table 11.** The Observable Cost of Rapidity for the Beijing–Shanghai High-speed Railway, $S_{13}$ (unit: CNY).

| City | Nanjing | Changzhou | Wuxi | Shanghai |
|---|---|---|---|---|
| Beijing | 519.4 | 530.9 | 613.9 | 653.8 |
| Tianjin | 379.92 | 381.63 | 468.88 | 488.74 |
| Jinan | 333.88 | 327.76 | 407.71 | 425.59 |

**Table 12.** The Observable Cost of Rapidity for Civil Aviation $S_{23}$ (unit: CNY).

| City | Nanjing | Changzhou | Wuxi | Shanghai |
|---|---|---|---|---|
| Beijing | 360.83 | 339.54 | 396.94 | 398.57 |
| Tianjin | N | N | N | 286.91 |
| Jinan | N | N | N | 283.5 |

- The Observable Cost of Convenience $S_{i4}$

To measure the observable cost of convenience, $S_{i4}$, we need to know the time for passengers to check in, $t_{i4}^4$, and the time to wait for boarding, $t_{i4}^5$. However, it is difficult to distinguish between $t_{i4}^4$ and $t_{i4}^5$ in reality. Therefore, in general, scholars often use questionnaires to sample the sum of $t_{i4}^4$ and $t_{i4}^5$. Wang [52] found that when taking a train, 80 percent of passengers choose to arrive at the station within 61 to 71 min before boarding; Brunetta [53] pointed out that when passengers choose to travel by civil aviation, more than half of passengers choose to arrive at the airport at least two hours in advance. In view of this situation, this paper assumes that when people choose to travel by high-speed railway, most people tend to arrive at the station 1.1 h earlier; when they choose to take an airplane, most people tend to arrive at the airport 2 h earlier. In combination with the results in Table 10, the calculating results of $S_{i4}$ are shown in Tables 13 and 14.

**Table 13.** The Observable Cost of Convenience for the Beijing–Shanghai High-speed Railway, $S_{14}$ (unit: CNY).

| City | Nanjing | Changzhou | Wuxi | Shanghai |
|---|---|---|---|---|
| Beijing | 88.99 | 82.27 | 92.9 | 88.57 |
| Tianjin | 68.74 | 62.01 | 72.64 | 68.31 |
| Jinan | 74.5 | 67.77 | 78.4 | 74.07 |

**Table 14.** The Observable Cost of Convenience for Civil Aviation $S_{24}$ (unit: CNY).

| City | Nanjing | Changzhou | Wuxi | Shanghai |
|---|---|---|---|---|
| Beijing | 161.81 | 149.58 | 168.91 | 161.03 |
| Tianjin | N | N | N | 124.2 |
| Jinan | N | N | N | 134.68 |

- The Observable Cost of Comfort $S_{i5}$

According to the research of Zhang and Peng [54], the numerical values of the parameters involved in calculating the observable cost of comfort are shown in Table 15.

**Table 15.** The Numerical Values of the Parameters.

| | High-Speed Railways | Civil Aviation |
|---|---|---|
| Time for Passengers to Eliminate Fatigue ($H$) | 15 | 15 |
| Undetermined coefficient ($\alpha_i$) | 59 | 79 |
| Undetermined coefficient ($\beta_i$) | 0.29 | 0.25 |

According to the Equation (9) and the results in the related tables, the calculated results of the observable cost of comfort, $S_{i5}$, are shown in Tables 16 and 17.

**Table 16.** The Observable Cost of Comfort for the Beijing–Shanghai High-speed Railway (unit: CNY).

| City | Nanjing | Changzhou | Wuxi | Shanghai |
|------|---------|-----------|------|----------|
| Beijing | 158.34 | 173.35 | 204.06 | 238.2 |
| Tianjin | 112.19 | 120.43 | 153.07 | 173.26 |
| Jinan | 90.17 | 90.89 | 116.67 | 128.51 |

**Table 17.** The Observable Cost of Comfort for Civil Aviation $S_{25}$ (unit: CNY).

| City | Nanjing | Changzhou | Wuxi | Shanghai |
|------|---------|-----------|------|----------|
| Beijing | 72.61 | 68.4 | 80.19 | 81.06 |
| Tianjin | N | N | N | 57.87 |
| Jinan | N | N | N | 56.99 |

Thus far, the observable costs for the Beijing–Shanghai High-speed railway and civil aviation have all been calculated.

4.1.3. The Calculation of the Passenger Flow Sharing Rate

Based on the calculated costs, the passenger flow sharing rate for the Beijing–Shanghai High-speed Railway and civil aviation can be measured and calculated using Equations (2)–(4). Based on the results of the questionnaire distributed in the preliminary investigation, this paper determines the weights of each cost, $\mu_1$, $\mu_2$, $\mu_3$, and $\mu_4$, in Equation (4) to take 0.555, 0.295, 0.05 and 0.1, respectively. Therefore, the calculated results of the observable cost, $R_i$, are shown in Tables 18 and 19.

**Table 18.** The Observable Cost for the Beijing–Shanghai High-speed Railway, $R_1$ (unit: CNY).

| City | Nanjing | Changzhou | Wuxi | Shanghai |
|------|---------|-----------|------|----------|
| Beijing | 431.03 | 462.81 | 503.11 | 571.33 |
| Tianjin | 361.5 | 393.02 | 428.1 | 515.13 |
| Jinan | 265.53 | 300.64 | 341.21 | 373.54 |

**Table 19.** The Observable Cost of Civil Aviation, $R_2$ (unit: CNY).

| City | Nanjing | Changzhou | Wuxi | Shanghai |
|------|---------|-----------|------|----------|
| Beijing | 554.7 | 555.71 | 585.89 | 616.59 |
| Tianjin | N | N | N | 568.39 |
| Jinan | N | N | N | 479.02 |

After obtaining the observable costs, $R_1$ and $R_2$, the average observable costs can be calculated using Equation (3), and the passenger flow sharing rate, $P_i$, can be calculated using Equation (2). The results are shown in Table 20.

**Table 20.** The Passenger Flow Sharing Rate for the Beijing–Shanghai High-speed Railway, $P_1$, and Civil Aviation, $P_2$.

| | | Nanjing | Changzhou | Wuxi | Shanghai |
|------|------|---------|-----------|------|----------|
| Beijing | $P_1$ | 56.24% | 54.55% | 53.79% | 51.9% |
| | $P_2$ | 43.76% | 45.45% | 46.21% | 48.1% |
| Tianjin | $P_1$ | 1 | 1 | 1 | 52.46% |
| | $P_2$ | N | N | N | 47.54% |
| Jinan | $P_1$ | 1 | 1 | 1 | 56.15% |
| | $P_2$ | N | N | N | 43.85% |

It is not difficult to see from the data in the table that when considering the competition between only high-speed railways and civil aviation, most people's choice of the modes of transportation conforms to the rule that when the distance between departure A and destination B is short, people are more willing to travel by high-speed railways, which are faster and more convenient; as the mileage between the two places expands, people are more inclined to choose civil aviation, which has a shorter operating time and a provides good comfort.

### 4.2. The Calculation of the Optimal Ticket Price for the Beijing–Shanghai High-Speed Railway

From the Equation (16), we find that when calculating the optimal ticket price, it is necessary to estimate the passenger volume under the current price, $q_1$, the new observable cost after the ticket price changed, $R_1'$, and the operating cost, $c_1$. Beijing and Shanghai are two megacities of China, and the traffic is convenient and frequent between them; thus, the competition between high-speed railways and civil aviation is fierce. In addition, the annual passenger volume of civil aviation from Beijing to Shanghai is more accurately counted. Based on these facts, this paper only uses the route from Beijing to Shanghai as an example for calculating the optimal ticket price for high-speed railways and explores the mechanism of the optimal ticket pricing for high-speed railways.

According to news reports, in 2019, the annual passenger volume of civil aviation from Beijing to Shanghai was 8123,735, and the daily passenger volume was approximately 22,256. Considering the passenger volume of the high-speed railway from Beijing to Shanghai is difficult to obtain directly, this paper uses the passenger flow sharing rate calculated above to indirectly obtain the passenger volume of the high-speed railway from Beijing to Shanghai. The calculating process is as follows:

$$q_1 = \frac{q_2}{P_2} \times P_1 = \frac{22,256}{48.1\%} \times 51.9\% \approx 24,014 \tag{19}$$

According to Equation (4), there will be a new observable cost for the Beijing–Shanghai High-speed Railway, $R_1'$, when the ticket price changes. The calculation of $R_1'$ is shown as follows:

$$R_1' = 1 \times (0.555 \times p' + 0.295 \times 653.82 + 0.05 \times 88.57 + 0.1 \times 238.2) = 0.555p' + 221.13 \tag{20}$$

As for the numerical value of the operating cost, $c_1$, data from the public interview show that the person–kilometer operating cost of the Beijing–Shanghai High-speed Railway is about CNY 0.45 to 0.5. This paper takes the median value, 0.475, as the numerical value of $c_1$. It is also known that 22 pairs of EMU trains depart daily from Beijing to Shanghai, and the mileage for this route on the Beijing–Shanghai High-speed Railway is 1318 km. Thus, the operating cost can be calculated as follows:

$$c_1 = 0.475 \times 1318 \times 22 \times 2 = 27,546.2 \tag{21}$$

Equation (17) is a constraint on the range of price fluctuations. The latest announcement from the Beijing–Shanghai High-speed Railway pointed out that the lowest price for a second-class seat from Beijing to Shanghai was CNY 498. Therefore, the optimal ticket price, $p_1'$, should meet the constraint condition like this:

$$p_1' \geq 498 \tag{22}$$

Equation (18) is the constraint condition on the operating income. The current ticket price for a second-class seat from Beijing to Shanghai is CNY 631. Combining this with the calculation result of Equation (19), it can be seen that before the ticket price changed, the operating income for second-class seats from Beijing to Shanghai is:

$$I = 24,014 \times 631 = 15,152,834 \tag{23}$$

Therefore, the new operating income, $I'$, should be no less than CNY 15,152,834.

In summary, the optimal pricing model for the Beijing–Shanghai High-speed Railway can be expressed as follows:

$$\max(\pi) = p_1' \times 24,014 \times \left[ \left( \frac{571.33}{0.555p_1' + 221.13} \right)^{0.709} + \frac{e^{-\frac{0.555p_1' + 221.13}{0.2775p_1' + 396.23}}}{0.519} - 1 \right] - 27,546.2 \quad (24)$$

$$\text{s.t.} p_1' \geq 498 \quad (25)$$

$$p_1' \times 24,014 \times \left[ \left( \frac{571.33}{0.555p_1' + 221.13} \right)^{0.709} + \frac{e^{-\frac{0.555p_1' + 221.13}{0.2775p_1' + 396.23}}}{e^{-\frac{0.555p_1' + 221.13}{0.2775p_1' + 396.23}} + e^{-\frac{571.33}{0.2775p_1' + 396.23}}} - 1 \right] \geq 15,152,834 \quad (26)$$

Solving for Equation (24), it is determined that when $p_1'$ is about 850, the enterprise can obtain the largest profit. At this time, the daily operating income of the enterprise for transporting passengers between Beijing and Shanghai is CNY 15,170,424.36, and the daily passenger volume generated is approximately 17,847.

### 4.3. Analysis of the Impact of Ticket Price on Financial Sustainability

In the aforementioned case, the changes in operating income, profits, and passenger volume caused by the change of the ticket price are shown in Table 21.

**Table 21.** Table of Changes in Ticket Price, Operating Income, Profits, and Passenger Volume.

|  | Current Ticket Price ($p_1 = 631$) | Optimal Ticket Price ($p_1' = 850$) | Variation |
|---|---|---|---|
| Operating Income | 15,152,834 | 15,170,424.36 | 17,590.36 |
| Operating Cost | 27,546.2 | 27,546.2 | — |
| Profits from Operating Activities | 15,125,287.8 | 15,142,878.16 | 17,590.36 |
| Passenger Volume | 24,014 | 17,847 | −6167 |

The analysis of sensitivity plays an important role in financial management and forecasting. According to the data in Table 21, the price elasticity of demand for the Beijing–Shanghai High-speed railway is:

$$E_d = -\frac{-6167}{850 - 631} \times \frac{631}{24,014} = 0.74 \quad (27)$$

It can be seen that high-speed railways are inelastic. Therefore, for high-speed railroad transportation enterprises, appropriately increasing the ticket price within a reasonable range is conducive to increasing operating income, thereby enhancing the profitability, reducing liquidity risks, and promoting the sound and sustainable development of transportation enterprises. The impact mechanism of the ticket price on financial sustainability can be shown in Figure 2.

The figure shows that the increase in ticket price can lead to an increase in the operating income and cash flow, which is well reflected in Table 21. Thus, the indices related to the sustainable operation, growth margin and operating profit margin, will increase, and the ratio of debt repayment and the current ratio related to sustainable debt repayment have the same change. The change of the growth rate of operating income leads to a change in

earnings per share, which is one of the constituent parts of a sustainable growth rate. Thus, the increase in operating income brings the realization of sustainable development, which is a positive signal for investors that the operating activities are running well, and more funds will be invested to expand the scale of business, which in turn, promotes an increase in operating income and cash flow.

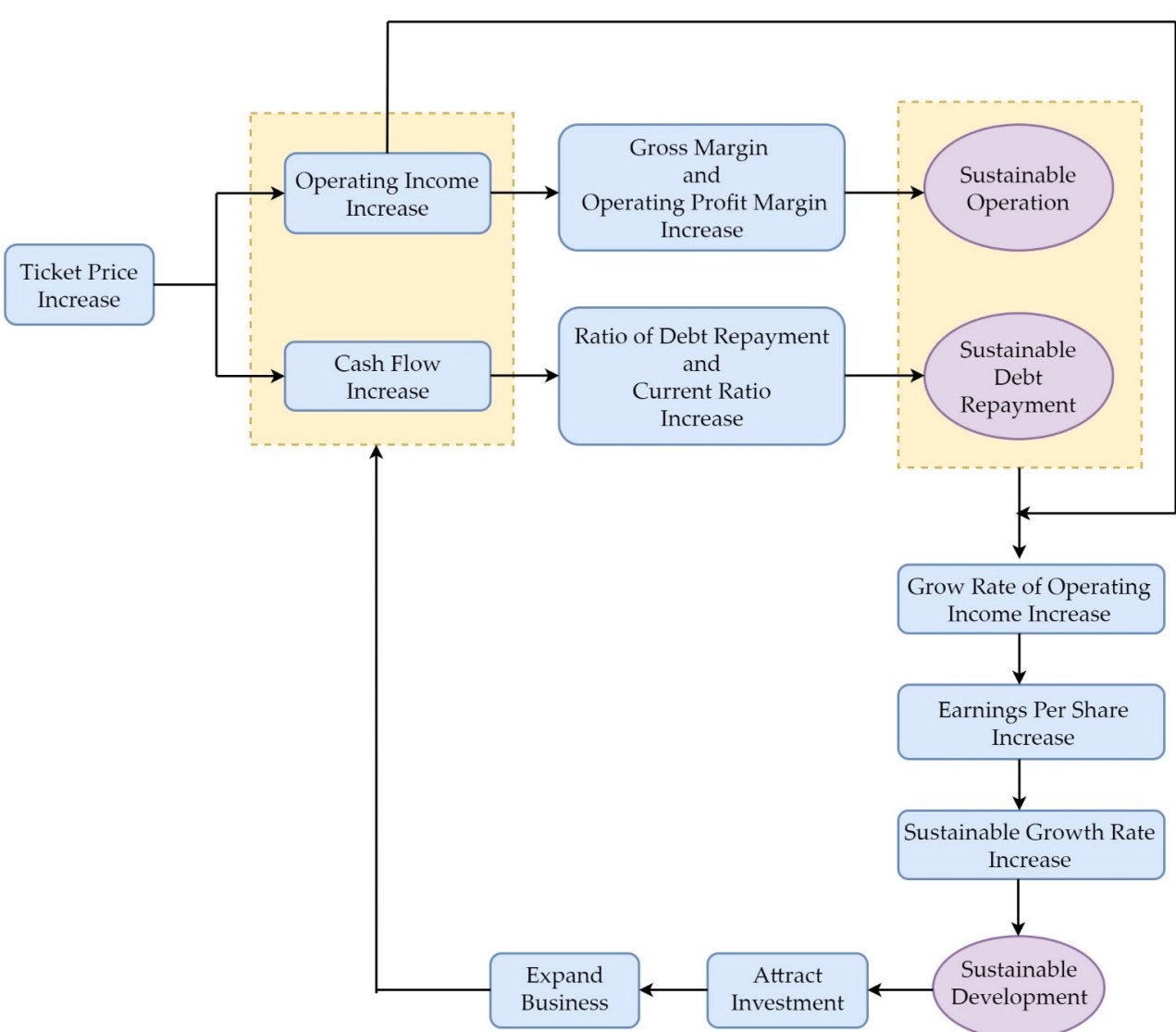

**Figure 2.** The Impact Mechanism of Ticket Price on Financial Sustainability.

From the perspective of whether the ticket price covers the operating cost, the ticket price rates for the Beijing–Shanghai High-speed Railway under circumstances of $p_1$ and $p_1'$ are:

$$p = p_1: \ traffic \ price \ rate = \frac{631}{1318} = 0.479 \tag{28}$$

$$p = p_1': \ traffic \ price \ rate = \frac{850}{1318} = 0.645 \tag{29}$$

After comparing with the operating cost, $c_1$, it is not difficult to find that if $c_1$ takes the maximum value of CNY 0.5, then the current ticket price, $p_1$, cannot cover the operating cost at all. After deducting the huge depreciation costs and interest expenses, the net profit will be negative. This is obviously not a good state of financial sustainability. When the optimal ticket price, $p_1'$, is implemented, not only will $c_1$ be covered, but the enterprise will

also make a profit of at least CNY 0.145 per person–kilometer. Based on the results listed in Table 21, the profits generated from the operating activities can be calculated as follows:

$$profits\ generated\ from\ operating\ activities = 0.145 \times 1318 \times 17,847 \times 365 = 1,244,920,162.05 \tag{30}$$

According to the relevant financial information disclosed in the Annual Report of the Beijing–Shanghai High-speed Railway Co., Ltd. (hereafter called "Annual Report"), in 2019, the interest expenses that the enterprise should pay were CNY 1,099,497,716.7. When the ticket price reaches the optimal state, the operating income obtained can not only make up for the operating costs, but also cover the interest expenses. This is a favorable guarantee for the enterprise to be able to obtain a stable income, alleviate the pressure on debt repayment, and realize sustainable operation and debt repayment.

Unlike enterprises in other industries, transportation enterprises can obtain cash flow as soon as the sales are completed, since there are no accounts receivable in transportation enterprises. Therefore, the change in the ticket price not only affects the current operating income but also has an immediate manifestation in current cash flow. From the operational perspective, the increase in operating income has brought about an increase in gross profit margin. Regardless of other non-operating expenses, the operating profit margin will also change positively. According to the data in Table 21, in this case, the average daily gross profit margin can be calculated as follows:

$$p = p_1:\ GM_1 = \frac{15,125,287.8}{15,152,834} \times 100\% = 99.8182\% \tag{31}$$

$$p = p_1':\ GM_1' = \frac{15,142,878}{15,170,424.36} \times 100\% = 99.8184\% \tag{32}$$

Although the daily gross profit margin has only increased by 0.002‰, from the perspective of the whole year, the growth will be quite considerable.

According to the 2019 annual report, the total sales expenses, general expenses, and management expenses of the enterprise were CNY 269,179,120.89, and the passenger turnover quantity was 956,100,000,000 person–kilometers. On average, the amortized cost per person–kilometer was:

$$the\ amortized\ cost\ per\ person - kilometer = \frac{269,179,120.89}{956,100,000,000} = 0.00028 \tag{33}$$

In order to calculate and illustrate the change in the operating profit margin caused by the change in the ticket price, this paper sets $c_1$ as equal to 0.475, which was used in the calculation above, and the operating profit margin can be calculated as follows:

$$p = p_1:\ the\ operating\ profit\ margin = \frac{\frac{631}{1318} - 0.475 - 0.00028}{\frac{631}{1318}} = 0.0078 \tag{34}$$

$$p = p_1':\ the\ operating\ profit\ margin = \frac{\frac{850}{1318} - 0.475 - 0.00028}{\frac{850}{1318}} = 0.2631 \tag{35}$$

The optimization of ticket prices has brought about an increase in the operating profit margin, and this growth is very obvious. Therefore, from the perspective of operation sustainability, the optimal ticket price plays a powerful role in promoting normal operating activities and realizing the long-term and healthy development of the enterprises.

High-speed railway projects of have a large scale of investment and a high required rate of return, which has brought considerable debt repayment pressure to transportation enterprises. From the perspective of debt servicing, the growth of income causes an increase in cash flow. When the total debt and the total current debt are fixed, the current ratio and the ratio of debt repayment will increase; thus, the solvency of the enterprises will be enhanced. According to the annual report, in 2019, the total liabilities of the enterprise were CNY 26,377,171,213.72, the current liabilities were CNY 5,494,278,431.16, and the passenger

turnover quantity was 956,100,000,000 person–kilometers. Due to the lack of accurate data, this paper uses total liabilities, total current liabilities, and passenger turnover quantity to roughly estimate the total debt and total current liabilities per person–kilometer:

$$total\ debt\ per\ person-kilometer = \frac{26,377,171,213.72}{956,100,000,000} = 0.0276 \tag{36}$$

$$total\ current\ debt\ per\ person-kilometer = \frac{5,494,278,431.16}{956,100,000,000} = 0.0057 \tag{37}$$

Combining the data in Table 21, with set $c_1$ equal to 0.475, the current ratio and ratio of debt repayment can be calculated as follows:

$$p = p_1 : ratio\ of\ debt\ repayment = \frac{\frac{631}{1318} - 0.475}{0.0276} = 0.145 \tag{38}$$

$$p = p_1' : ratio\ of\ debt\ repayment = \frac{\frac{850}{1318} - 0.475}{0.0276} = 6.16 \tag{39}$$

$$p = p_1 : current\ ratio = \frac{\frac{631}{1318}}{0.0057} = 84.04 \tag{40}$$

$$p = p_1' : current\ ratio = \frac{\frac{850}{1318}}{0.0057} = 113.18 \tag{41}$$

By comparing the results, it can be seen that in the case of this paper, the optimal ticket price leads to an increase in the two ratios, and the effect of financial leverage is significant. Thus, for transportation enterprises, the optimal ticket price plays an important role in realizing sustainable debt repayment.

In general, it often takes decades for a high-speed railway to go from construction to normal operation. Therefore, it is necessary for enterprises and investors to evaluate the sustainable development potential of high-speed railways. Operating income is the main source of profits for transportation enterprises, and is also a guarantee for the interests of investors. In the case of this paper, the growth rate of the operating income can be calculated as follows:

$$the\ growth\ rate\ of\ operating\ income = \frac{15,170,424.36 - 15,152,834}{15,152,834} \times 100\% = 0.116\% \tag{42}$$

It can be seen that after the ticket price had changed, the daily operating income of the enterprise achieved an increase of about 0.1%. Although this growth rate is not large, it only represents the growth rate of the operating income from the Beijing to Shanghai route in a single day. If we consider the annual operating income growth of the entire Beijing–Shanghai High-speed Railway, this growth rate is quite considerable. Stable growth of the operating income will bring about an increase in net profits, and with a certain number of shares issued, earnings per share will also increase. The earnings per share and the rate of profit reinvestment affect the sustainable growth rate, which is a manifestation of the sustainability of development. Therefore, the optimal ticket price also plays a key role in achieving the sustainable development of enterprises.

In summary, the impact of the ticket price on the financial sustainability of the Beijing–Shanghai High-speed Railway is shown in Table 22.

It can be seen from the data in the table that when the ticket price is optimized, the profitability, solvency, and sustainable development capabilities of the Beijing–Shanghai High-speed Railway have been significantly improved. By affecting the operating income and cash flow, the ticket price has a significant impact for transportation enterprises on the realization of the path of "sustainable operation-sustainable debt repayment-sustainable development".

**Table 22.** The Impact of the Ticket Price on the Financial Sustainability of the Beijing–Shanghai High-speed Railway.

| | Index | The Current Ticket Price $p_1$ | The Optimal Ticket Price $p_1{}'$ | Variation |
|---|---|---|---|---|
| Sustainable Operation | Gross Margin | 99.8182% | 99.8184% | 0.002‰ |
| | Operating Profit Margin | 0.0078 | 0.2631 | 0.2553 |
| Sustainable Debt Repayment | Ratio of Debt Repayment | 0.145 | 6.16 | 6.015 |
| | Current Ratio | 84.04 | 113.18 | 29.14 |
| Sustainable Development | Growth Rate of Operating Income | — | — | 0.116% |

## 5. Conclusions

Based on the perspective of financial sustainability, this paper explores the method for determining the ticket prices for high-speed railways, with full consideration of supply and demand, competition, and passenger's utility, and studies the impact of the ticket price on financial sustainability. Through analysis and calculation, we find that: firstly, the optimal ticket price of a second-class seat for the high-speed railway from Beijing to Shanghai is about CNY 850. Compared to the current ticket price, there is still a room for a price increase; secondly, the increase in the ticket price leads to the increase in operating income and cash flow, which affects the sustainability of the high-speed railway enterprises. When the ticket price changes from CNY 631 to CNY 850, the gross margin and operating profit margin increase by 0.002% and 0.2553%, respectively; the ratio of debt repayment and the current ratio increase by 6.015% and 29.14%, respectively. This kind of increase is a good reflection of sustainable operation and debt repayment. Besides, the change of the ticket price leads to an increase in the growth rate of the operating income by 0.116%, which can guarantee the realization of the sustainable development of the Beijing–Shanghai High-speed Railway.

The optimal pricing model for high-speed railways proposed in this paper provides a useful tool for enterprises to estimate the optimal ticket price under competitive conditions. Quantitative analysis of the financial sustainability of high-speed railways from the perspective of ticket price has enriched the research findings for financial sustainability.

The shortcomings of this paper are as follows: firstly, due to the difficulty of data acquisition, this paper only calculates the passenger flow sharing rate between several cities along the Beijing–Shanghai High-speed Railway, and only calculates the optimal ticket price from Beijing to Shanghai; similarly, due to a lack of some detailed data, such as the cover charge and energy bill of the EMU trains, the railway maintenance costs, etc., this paper only takes the total operating cost, $c_1$, into account when formulating the pricing model and does not specifically analyze the impact of each of these costs on ticket pricing. In addition, the Beijing–Shanghai High-speed Railway Co., Ltd. has only been listed for two years and was greatly affected by COVID-19 in 2020; the transportation industry has suffered significant damage. Therefore, the public data of transportation enterprises in 2020 is not universal. Thus, this paper takes only the relevant 2019 open public data for its research purposes. In summary, future research on relevant aspects is required to strengthen our understanding of pricing methods and the financial sustainability of high-speed railways.

**Author Contributions:** Conceptualization, X.H. and J.D.; methodology, R.L.; software, X.H.; validation, X.H., J.D. and R.L.; formal analysis, R.L.; investigation, X.H.; resources, X.H.; data curation, X.H.; writing—original draft preparation, X.H.; writing—review and editing, J.D.; visualization, R.L.; supervision, J.D.; project administration, R.L.; funding acquisition, X.H., J.D. and R.L. All authors have read and agreed to the published version of the manuscript.

**Funding:** This research was funded by the Fundamental Funds for Humanities and Social Sciences of Beijing Jiaotong University(2016JBWZ004).

**Institutional Review Board Statement:** Not applicable.

**Informed Consent Statement:** Not applicable.

**Data Availability Statement:** The data that support the findings of this study are available in the Annual Report of the Beijing–Shanghai High-speed Railway Co., Ltd., from some ticket-booking websites, and other news websites. These data were derived from the following resources available in the public domain: http://www.cr-jh.cn/websiteMenu/145/2 (accessed on 18 December 2021); http://www.ctrip.com/ (accessed on 18 December 2021); http://flight.qunar.com/ (accessed on 18 December 2021); http://www.12306.cn/ (accessed on 18 December 2021); http://www.sohu.com/ (accessed on 18 December 2021); http://www.phb123.com/ (accessed on 18 December 2021); http://www.gov.cn/ (accessed on 18 December 2021).

**Conflicts of Interest:** The authors declare no conflict of interest.

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
