# Peer review of "Research on High-Speed Railway Pricing and Financial Sustainability"

_sustainability, doi:10.3390/su14031239_

Round 1

Reviewer 1 Report

A fairly complete analysis of the formation of the cost of tickets for high-speed trains is given in the article. As a recommendation, for the future, it is advisable to consider other aspects of pricing and formation of the cost of tickets, such as reducing operating costs, reducing the cost of maintaining infrastructure, etc.

Reviewer 2 Report

This paper constructs a model of passenger flow sharing rate that reflects supply and demand, market competition and passenger utility, and applies it to the model of optimal ticket price of high-speed railways. And then, takes the listed Beijing-Shanghai high-speed railway as an example to calculate the optimal ticket price by this model, so as to provide a method which has a certain reference value for the pricing of high-speed railways. Using the optimal ticket price calculated, this study quantitatively analyzes the impact of ticket price changes on the financial sustainability of enterprises and explores the path for transportation enterprises of high-speed railways to achieve financial sustainability.

The topic is interesting, and the manuscript is very comprehensive and well written. The paper is very well documented, and references are comprehensive.

The results seem coherent and described with sufficient clarity. These look like the existing ticket price of Beijing to Shanghai is not the optimal one, and there is still room for price increasing. Also, the ticket price has an impact on financial sustainability by affecting corporate operating income and cash flow.

The main contributions of the paper are clearly highlighted, the study developed providing a method for formulating the optimal ticket price of high-speed railways. It also contributes to enrichment of the research on financial sustainability of high-speed railways.

It is noted that there are real prospects for research development by updating research data.

The text and English language in the paper must checked, edited, and corrected.

Reviewer 3 Report

The manuscript presents a study on the pricing and financial sustainability of a high-speed train system. I find the study is interesting and will be useful for readership.

I provide my comments below for the authors to consider for further improving the quality of the manuscript.

  • The original contribution and innovation of the current study should be clearly emphasised and presented in comparison with existing literature (e.g., studies published by Zhao 2017; Lin 2019; Jing et al. 2019)
  • I am not convinced how Equations 8 and 9 were derived. Please provide more information and the derivation procedure.
  • Table 1: I suggest including references/original sources for information and Equations presented in the Table.
  • Figure 2: I suggest including more information describing the process presented in the Figure.
  • Conclusion: I suggest to re-write the Conclusion section covering critical findings of the current study supported with actual numbers/values.

Round 2

Reviewer 3 Report

The authors have satisfactorily attended comments and suggestion provided by the Reviewer.